# Research on the generation of food safety network public opinion—Taking the Chinese Mouse Head Duck Neck Incident as an example

**Wei Yang, Jiayin Pei\*, Zheyu Lin**

School of Business, Jiangnan University, Wuxi, Jiangsu Province, China

\* 1956444384@qq.com

## Abstract

In recent years, the frequent occurrence of food safety incidents has posed a serious threat to people's lives and properties, and triggered heated discussions in the whole society. If the relevant departments shirk each other's responsibilities or deceive the public in handling the incident, it will further lead to the generation of network public opinion, which will have an impact on the stability of the society and the credibility of the government. In order to study the generation of food safety network public opinion, this article takes the sudden "mouse head and duck neck" incident in a university in Jiangxi Province, China as an example, and combines text mining with grounded theory. Firstly, the LDA topic clustering model is used to identify six main concerns (topics) of the public during the public opinion period, and the topic words and words with high TF-IDF values under each topic is counted. Based on this, the grounded theory method is used for three-level coding, and then a network public opinion generation model is constructed. It was found that the four main categories of national subjects, public's responses, value orientations, and news media play an important role in the generation process of network public opinion. These findings not only provide a reference for the governance of food safety public opinion in China, but also shed light on public opinion management in other countries, especially when responding to food safety incidents of general significance.

## Introduction

Food safety, as the cornerstone of public welfare, is intimately intertwined with people's daily lives and its significance is universally acknowledged worldwide. With economic progress and rising living standards, the food market has become increasingly diversified, presenting consumers with a broader array of options. Yet, this flourishing market also bears witness to the recurring presence of unhealthy and problematic foods, which silently escalate the risks associated with food safety hazards. In 2021, the UK's Food Standards Agency (FSA) reported a severe food safety incident,

**Data availability statement:** All data are available from the kaggle database (https://www.kaggle.com/datasets/zheyujndx/lzyfoodsafety).

**Funding:** National Social Science Foundation of China, Grant/Award Number: 20FTQB018; JiangSu Province Social Science Foundation, Grant/Award Number: 20TQB001; The Fundamental Research Funds for the Central Universities, Grant/Award Number: 2019JDZD06.

**Competing interests:** The authors have declared that no competing interests exist.

revealing that 480 cases of salmonella infections were traced back to contaminated meat products supplied by three distributors. This incident not only captured widespread public attention but also necessitated enhanced scrutiny of supply chains by relevant authorities. In the same year, a large-scale food poisoning event occurred at a Foxconn plant in India, causing severe physical distress among numerous employees, manifesting as dizziness, vomiting, and diarrhea. The gravity of the situation led to over 159 workers requiring hospitalization, compelling the factory to temporarily cease operations and undertake comprehensive food safety improvements. Turning to 2022, a major flaw in the production line of a U.S. infant formula manufacturer was uncovered, with the discovery of high concentrations of harmful bacteria in its products, leading to health complications for several infants and, tragically, reported infant fatalities. These incidents sparked a tense "formula shortage" across various regions in the U.S., prompting urgent government intervention and action to restore market order and protect the food safety of infants. Reflecting on 2023, a notable food safety controversy emerged at a university cafeteria in Guangdong Province, China, where a student discovered unusual gelatinous substances in their meals. The official explanation from the institution, identifying these substances as a normal component of duck meat—the eye membrane, failed to alleviate student discontent and instead ignited intense public backlash. Compared to other public health events, food safety incidents are more likely to resonate with the population, not only as a direct threat to individual health and lives, but also as a potential cause of mass illness [1]. Ensuring food safety, guarding people's health rights and maintaining social stability has become an urgent and important task.

The proliferation of the internet, particularly in China, has led to a substantial surge in the number of netizens, positioning the country as one of the world's largest online populations. According to the 53rd Statistical Report on Internet Development in China released by the China Internet Network Information Center (CNNIC), by December 2023, China's internet user base reached 1.092 billion, marking an increase of 24.8 million users since December 2022, with an internet penetration rate of 77.5%. The rise of the digital age has transformed the modes of information dissemination and opinion exchange, enabling citizens to transcend geographical and temporal barriers on social media platforms to freely discuss hot topics like food safety. These discussions often carry emotional tones and unique individual perspectives [2]. Given the diversity of public opinions and varied viewpoints, compounded by the interactive, convenient, and viral nature of social media platforms, cyberspace can swiftly accumulate negative sentiment and amplify misinformation, catalyzing the formation of network public opinion [3]. If not promptly and effectively managed, such online sentiments can instigate public panic, threaten social stability, and challenge the credibility of the government [4,5]. In June 2023, a food safety incident at a university canteen in Jiangxi Province, known as the "Mouse Head Duck Neck" incident, rapidly ignited online discourse, drawing extensive societal attention. Leveraging this case, this study integrates text mining with grounded theory to conduct a deep analysis of the generation of network public opinion surrounding food safety incidents. Given the massive scale of data involved, the traditional application of

grounded theory alone is limited. Therefore, this research innovatively employs a combination of Latent Dirichlet Allocation (LDA) topic modeling and Term Frequency-Inverse Document Frequency (TF-IDF) algorithms. First, according to the LDA thematic clustering model and TF-IDF algorithm to derive the thematic words under each topic and the words with higher TF-IDF values, and based on this, implement the three-level coding of the grounded theory to establish the food safety network public opinion generation model and analyze it.

Food safety incidents, like other public health incidents, penetrate deeply into all aspects of daily life and are closely related to people's interests and lives and health, often triggering heated public discussions. With the rapid development of social networks and the advent of the big data era, the frequency and intensity of people's online interactions are increasing, and this increasingly active online communication is becoming an important driving force for the formation and development of network public opinion. The resulting network public opinion has also become the focus of many scholars' research, prompting text mining to become an important method in network public opinion research [6,7]. Text mining is the process of extracting valuable information and knowledge from a large amount of unstructured text data, and it is usually combined with natural language processing techniques to realize the efficient use of text data [8,7]. Text mining has been widely used in network public opinion analysis. For example, some scholars used Python to preprocess multilingual and multi-source text data and proposed a method based on the spectral clustering algorithm, which combined with the LDA model to extract text topics, revealing the main content of people's discussions during the public opinion dissemination [2]. Some researchers combined the LDA topic model with the BERT model to provide an in-depth analysis of the people's concerns and sentiment classification during public opinion, which not only optimized the topic vectors, but also significantly improved the accuracy of sentiment classification, revealed the process of the public's emotional changes and the role of opinion leaders, and provided an important reference for understanding and guiding network public opinion [9,10]. Other studies have used LDA topic modeling and k-means to extract topics from text data about "Food Safety of Takeout" in Sina Weibo from 2015 to 2018, and found that the topics discussed by the public have gradually increased and diversified [11]. In addition, sentiment analysis is also an important part of text mining, which is of great significance in the field of network public opinion [12,13]. From sentiment lexicon to machine learning to deep learning, the techniques of sentiment analysis have been evolving, gradually evolving from simple rule matching to complex semantic understanding. For example, some researchers proposed a fine-grained short text sentiment analysis method based on machine learning, which increased the proportion and weight of sentiment words in feature words by introducing the N-CHI feature selection algorithm and the W-TF-IDF weight calculation algorithm, and significantly improved the classification accuracy [14]. Although text mining technology has demonstrated powerful data processing capabilities in network public opinion research, it still faces some limitations that cannot be ignored: this method often can only capture the superficial features of public opinion, and it is difficult to comprehensively and in-depth reflect the public's real concerns and emotional tendencies. Especially when facing complex and changing social phenomena, it is often difficult for text mining techniques to reveal the deep logic and dynamic evolution process behind network public opinion, thus limiting its comprehensiveness and accuracy in analyzing network public opinion [15,16].

Grounded theory, as a bottom-up qualitative research method that emphasizes the gradual construction of a theoretical framework through data analysis, has been widely used in the study of network public opinion [17]. For example, some scholars have conducted an in-depth exploration of Chinese netizens' risk perception and attitudes in the COVID-19 epidemic based on the grounded theory. By systematically analyzing massive online text data, including open coding, spindle coding and selective coding, a theoretical model reflecting COVID-19-related public opinions was successfully constructed. The model profoundly reveals the intrinsic mechanism of online opinion generation and provides a theoretical basis for understanding public attitudes toward public health emergencies [18]. Other scholars combined the 5W communication model and grounded theory to explore the role of social workers in epidemic prevention and control and their influence on public opinion generation by taking the hot articles and comments on the WeChat platform as the research object [19]. In addition, a study constructed a basic framework of network public opinion in the epidemic area by collecting

hot news comments on Weibo during the Shanghai epidemic and using qualitative analysis, grounded theory and WSR methodology. The study identified three core categories, 15 major categories, and 86 influencing factors, established the WSR framework of network public opinion elements, and revealed the development of network public opinion under the COVID-19 epidemic, which is of great theoretical value in guiding the public to form reasonable behaviors [20]. Another study used Python and grounded theory to analyze the evolution of network public opinion during major sports events, and found that public opinion went through three phases of "outbreak, repetition, and decline", which were influenced by emotional resonance, accident review, and extended reflection, and provided important references for response and governance [21]. However, grounded theory also has some limitations. First, traditional grounded theory is usually applicable to the analysis of small-scale data, and it is difficult to effectively deal with large-scale social media data; second, the coding process of grounded theory is highly dependent on the subjective judgment of the researcher, which may lead to biased research results. These shortcomings limit its application in the big data environment to some extent [22,23].

At the same time, some scholars have conducted in-depth studies on the same food safety incident (such as the "Mouse head Duck Neck" incident) and put forward a comprehensive framework of the "survival cycle" of public opinion dissemination, analyzing in detail the various stages of public opinion dissemination and proposing targeted emergency countermeasures. The framework analyzes in detail the various stages of public opinion dissemination and puts forward targeted emergency responses. Some other researchers have used advanced NLP methods (e.g., BERT-TextCNN and BERTopic) and network analysis techniques to explore the patterns of emotion dynamics and information dissemination [24,25]. However, their studies lack in-depth exploration of online opinion generation and fail to comprehensively reveal the core logic of opinion generation.

In summary, although text mining and grounded theory have their own advantages in public opinion analysis and have been widely used, most studies currently use a single method (e.g., text mining or grounded theory), and few scholars combine the two. This makes it difficult to balance large-scale data processing and deep logic mining, and to fully capture the complexity and dynamics of public opinion generation. In this paper, we try to combine text mining with grounded theory: first, we use LDA topic model and TF-IDF algorithm to conduct text mining on microblog text data to find out the main concerns of the public during the period of public opinion, and then, we use grounded theory to code and build a network opinion generation model based on the results of text mining to analyze the results, which will provide references to the relevant departments for network opinion management and provide new ideas for future research.

## Method

This paper employs the TF-IDF algorithm and Latent Dirichlet Allocation (LDA) topic modeling to extract key phrases from the text. Building upon this foundation, it utilizes Grounded Theory to construct a model for generating network public opinion on food safety incidents.

TF-IDF (Term Frequency-Inverse Document Frequency) is a pivotal component in text mining, serving as a prevalent statistical method and weighting technique for evaluating the significance of a term within a document or a corpus. The importance of a term escalates with its frequency within a document but diminishes as it recurrently appears across the entire corpus, thereby highlighting principal terms while suppressing less salient ones. The computational formula for TF-IDF is illustrated as Equation (1):

$$TF\text{-}IDF = TF \times IDF = \frac{n}{N} \times \log(\frac{D}{d+1})$$

(1)

Where n represents the probability of words appearing in a certain document; N represents the probability of words appearing in the entire corpus; D represents the number of documents in the corpus; D represents the number of documents where the word is located (d + 1 prevents the denominator from being 0).

The number of topics is an important parameter of LDA topic modeling, which is usually determined by perplexity, topic consistency and other methods in topic mining. Compared with other methods, perplexity is a long-used metric in the field of machine learning, especially in the field of topic modeling, which has a strong generalization ability to unseen text data, and at the same time can automate the selection of the number of topics, thus reducing manual interference and making the selection process more objective. Therefore, this paper finally chooses to use the confusion degree to determine the number of topics.

Perplexity is a concept in information theory used to measure how well a probabilistic model predicts samples. In LDA modeling, a lower perplexity usually means that the model fits the data better. The specific calculation process can be expressed in equation (2):

$$\text{Perplexity (D)} = \exp\{\frac{-\sum_{d=1}^{M}\log(p(Wd))}{\sum_{d=1}^{M}Nd}\}$$

(2)

Among them, D represents the collection of documents in the corpus, with a total of M articles; $N_d$ represents the number of words in each document; Wd represents words in the document; P ($W_d$) represents the probability of the word $W_d$ appearing. This article selects the number of topics with lower confusion and relatively fewer topics as the parameters of the LDA topic model.

The LDA (Latent Dirichlet Allocation) topic model is a three-layer Bayesian model that includes three levels: document, topic, and word [26]. The LDA topic model converts documents in the word vector space into topics through dimensionality reduction, which is an important topic mining method in the field of natural language processing. The specific steps for document generation are shown below, and a diagram of the generation process is shown in Fig 1:

(1) From α The topic polynomial distribution for generating document i in the Dirichlet distribution of parameters $\theta_i$.

(2) Generate the topic $Z_{i,j}$ corresponding to the j-th word from the topic polynomial $\theta_i$ of the document.

(3) From β Generate word distribution $\varphi_{Zi,j}$ corresponding to topic $Z_{i,j}$ in the Dirichlet distribution of the parameter.

(4) Generate the final word $W_{i,j}$ from the polynomial distribution $\varphi_{Zi,j}$ of the word.

(5) Repeat the above four steps until a complete document is generated.

Where α The prior parameter of the Dirichlet distribution representing the topic distribution of each document, β The prior parameter of the Dirichlet distribution representing the distribution of each topic word, θ For the document topic

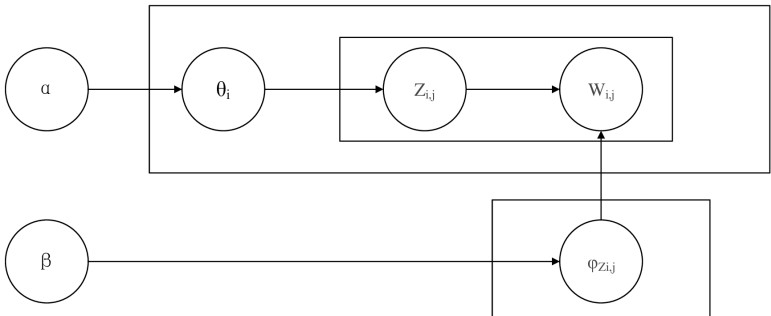

**Fig 1. LDA model structure diagram.**

distribution matrix, φ Represents the topic word distribution matrix, where $Z_{i,j}$ is the topic generated from the topic polynomial $\theta_i$ and $W_{i,j}$ is the word generated from the word polynomial $\varphi_{Zi,j}$.

By backward reasoning the above document generation process, the potential document-topic distribution θ and topic-word distribution φ can be deduced from the existing document data.

Grounded theory, introduced by Glaser and Strauss in 1968, is a qualitative research method that builds substantive theories from the ground up. It involves analyzing and extracting raw text data, identifying core categories, and refining them through continuous modification to construct a social theory [27]. The process primarily consists of five steps: data collection, open coding, axial coding, selective coding, and saturation testing. Open coding begins with conceptualizing and categorizing the raw text data. Axial coding then refines these initial categories by grouping those with similar attributes, resulting in more concise main categories. Selective coding integrates these main categories, establishes models to analyze their relationships, and gradually forms a coherent theory. Finally, saturation testing involves collecting new data, coding it, and checking for the emergence of new categories to validate the model's completeness. Its flowchart is shown in Fig 2

## Results and discussion

In order to study the generation of network public opinion on food safety incidents, this paper takes the food safety incident of "Mouse Head Duck Neck" that happened in the cafeteria of a university in Jiangxi Province in June 2023 as an example, and crawls the microblogs and their comments on Sina Weibo using web crawler technology, with a total of 24,588 records (Data source: lzyfoodsafety). Some of the data are shown in Table 1. These data include user name, date

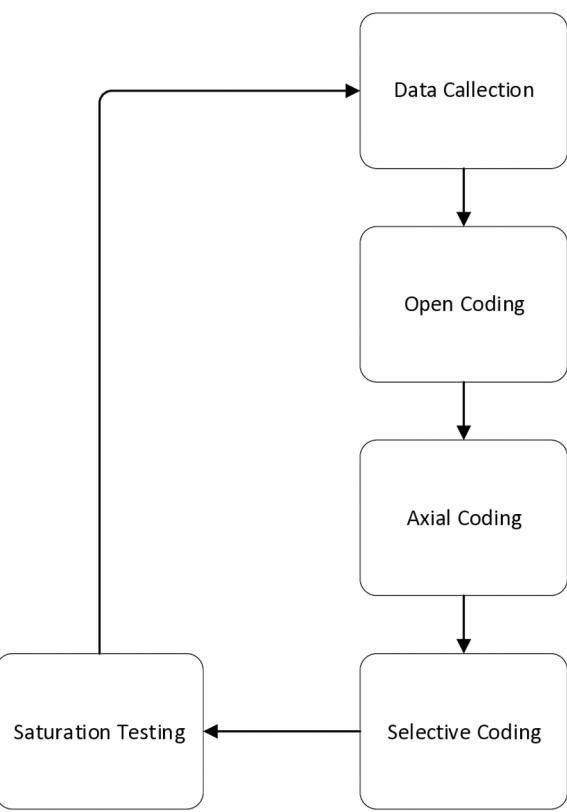

**Fig 2. Grounded theory.**

**Table 1. Weibo Text Example of the 'Mouse Head Duck Neck' Incident.**

| User | Comment Time | Comment Time |
|------|--------------|--------------|
| Cheryl's Hairpin | 2023-06-22 11:45:27 | Food safety really deserves strong regulation. It is absolutely beneficial to the country and the people. Without being overly critical, hygiene and safety are essential. Everyone, thank you for your hard work. |
| Nanchang Xiong Ming | 2023-06-21 23:00:08 | Food safety incidents in schools are happening frequently. All regions should take this as a warning. |
| China Liu Jie | 2023-06-17 15:03:09 | The so-called "post-truth" is just artificially obstructing the truth. If the media cannot uncover the truth, and regulatory departments cannot promptly reveal the truth, then the inversion of right and wrong becomes the so-called coated truth. |
| Empty Daoist 7 | 2023-06-17 14:55:36 | A modern version of "calling a deer a horse". |

of publication, content of the tweets, etc., which show the dissatisfaction of microbloggers about the food safety incident, their worries about the current situation of food safety, and their discussions about the relevant authorities. Specifically, we use a crawler to automatically access the target microblog page and extract the relevant microblogs and their comments, search by setting the keyword "mouse head duck neck", and store the acquired data in an Excel table. In the process of data collection and analysis, we strictly followed the terms and conditions of Sina Weibo, and only captured publicly accessible microblogs and their comments, without involving any private or sensitive information, to ensure the compliance and ethical use of data collection and utilization.

Since some of the crawled data contains invalid information such as emoticons, repeated text comments, and various miscellaneous symbols in the text, it cannot be used directly, and needs to be preprocessed before use. Although emoticons in social media posts can convey rich emotional nuances, we chose to remove them to simplify the text analysis process and avoid interference with the calculation of topic words and TF-IDF values. Given that the primary focus of this study is on topic analysis and the generation mechanism of network public opinion, rather than sentiment analysis, the removal of emoticons is not expected to significantly impact the core findings of the research. re.sub() is a function provided by the re module in python, which replaces the content in a string with a regular expression. In this article, we first use re.sub() function to match the emoticons and various miscellaneous symbols in the text data and replace them with spaces. After completing this step, the next step is to split the text, this paper uses the jieba library in python to do this step, this experiment uses the jieba library in python and regular expressions to split the text, de-dupe words and remove emoticons, etc. The jieba library is a third-party thesaurus for Chinese word splitting, and it can find out the maximum cut combinations based on the frequency of the words through dynamic programming and directed acyclic figures, etc. The maximum cut combinations of the words can be found through the Hierarchical Hierarchical Hierarchical Hierarchy (HHH). jieba library is a third-party lexicon for Chinese word segmentation. It finds the maximum cut combination based on word frequency through dynamic programming and directed acyclic figure, and predicts the unregistered words through HMM model, which can effectively improve the precision of word segmentation. After the word segmentation, we need to deactivate the words, due to the text mining process will appear some proper nouns, in order to improve the accuracy of the text mining results, can't directly use the existing deactivation word list, need to be combined with the proper nouns in the text data to create a customized deactivation word list and use it. In this paper, we use the jieba library to segment the text of the crawled microblogs, and then combine it with the customized deactivation word list based on the deactivation word list of the Harvard University to remove the deactivated words in the text that do not have any practical significance, and then remove the emoticons and all kinds of symbols through the regular expression. After applying the above processing steps to the crawled data, a total of 19,450 pre-processed microblog texts were obtained through excel de-weighting.

Manual merging of synonyms for words ranked in the top 100 in terms of word frequency after word splitting, for example, placing "college", "university," and "school" under the term "school", one can use a word cloud to confirm how effective the segmentation was. The important terms in the event are visually represented by the word cloud, where more frequency terms are indicated by larger font sizes. Fig 3 displays the word cloud related to this food safety issue. Fig 3 makes it clear that the word "mouse" is used the most in relation to this incident, suggesting that the public is quite alarmed by the "mouse head duck neck" incident that happened in the school. Furthermore, words like "canteen," "foreign body", "student" and "school" are used numerous times, indicating that the public is aware of and concerned about issues related to food safety on campuses.

The significance of words in the text can be inferred from the size of TF-IDF values. Fig 4 illustrates how the top 10 terms in this study were ordered using their TF-IDF values following segmentation. Words like "canteen", "mouse", "foreign body", "student" etc., from Fig 4 show what the public and major media sources were primarily focused on following the event.

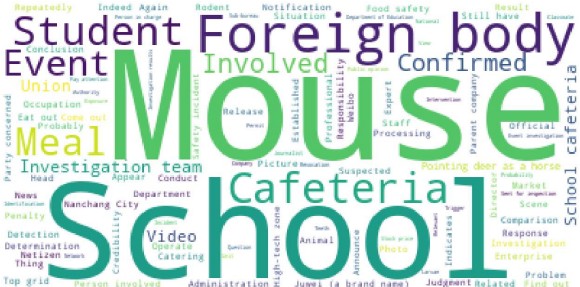

**Fig 3. Word Cloud.**

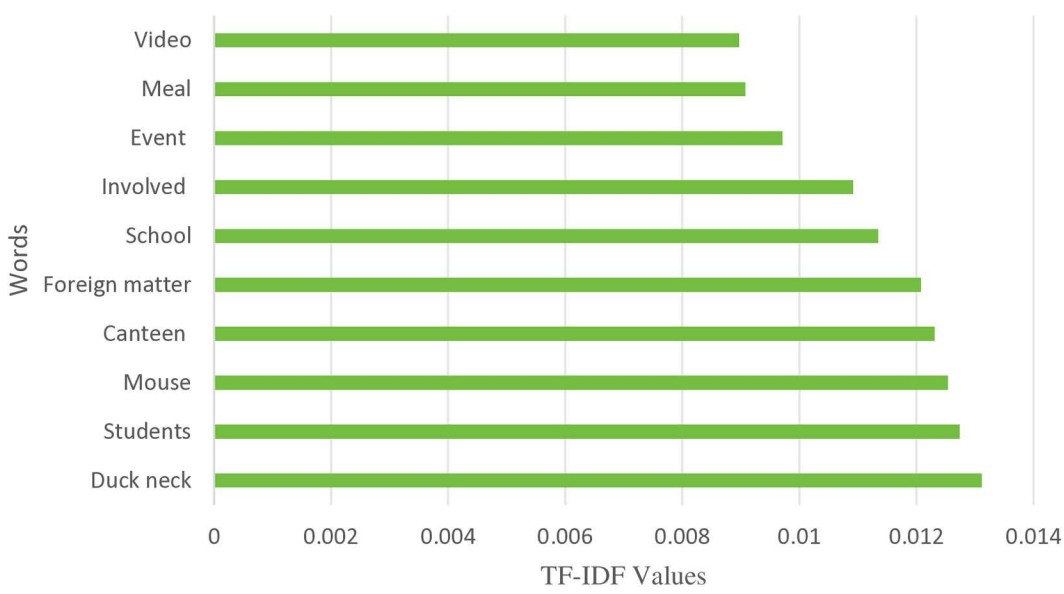

**Fig 4. TF-IDF Value Chart.**

**Table 2. Co-word Matrix.**

| | Duck neck | Mouse | School | Foreign body | Canteen | Student | Meal | Involved | Event | Video |
|---|---|---|---|---|---|---|---|---|---|---|
| Duck neck | | 24706 | 18581 | 20854 | 16373 | 17209 | 12505 | 12542 | 13634 | 11132 |
| Mouse | 24706 | | 9255 | 9449 | 10314 | 10116 | 6915 | 4506 | 7386 | 64395 |
| School | 18581 | 9255 | | 7757 | 8867 | 9364 | 6267 | 4986 | 5579 | 3535 |
| Foreign body | 20854 | 9449 | 7757 | | 14035 | 13881 | 8729 | 10172 | 5337 | 9883 |
| Canteen | 16373 | 10314 | 8867 | 14035 | | 17760 | 3778 | 12251 | 5242 | 9280 |
| Student | 17209 | 10116 | 9364 | 13881 | 17760 | | 2714 | 9117 | 5282 | 10282 |
| Meal | 12505 | 6915 | 6267 | 8729 | 3778 | 2714 | | 2183 | 2207 | 2734 |
| Involved | 12542 | 4506 | 4986 | 10172 | 12251 | 9117 | 2183 | | 4603 | 6151 |
| Event | 13634 | 7386 | 5579 | 5337 | 5242 | 5282 | 2207 | 4603 | | 2724 |
| Video | 11132 | 6495 | 3535 | 9883 | 9280 | 10282 | 2734 | 6151 | 2724 | |

Based on the tokenized microblog texts, a co-occurrence matrix is created, and Table 2 displays the top 10 co-occurrences. The co-occurrence matrix is then used to create a semantic relationship network figure, as illustrated in Fig 5. The size of the node represents the number of times the word appears in all the text data, the line at the node represents the node word and another node word co-occur in the same text data; the thickness of the line between the nodes represents the number of times the two node words co-occur in the same text data.

Terms like "school", "canteen", "mouse", "foreign body", "investigation team" and "truth" co-occur frequently, according to the semantic relationship network figure of important terms, suggesting that Weibo users are paying close attention to this campus food safety event. People are curious to learn the real story. Words with high co-occurrence rates, such as "involved", "punishment", "enterprise", "administration" and "supervision" indicate that Weibo users are keenly watching to see if the school and the cafeteria contractor will face consequences from the appropriate authorities. Weibo users' doubts and dissatisfaction regarding the credibility of the relevant authorities have been raised by the investigation team's preliminary handling results, as evidenced by the frequent co-occurrence of words like "investigation team", "public opinion", "credibility", "responsibility" and "seek truth from facts". Online users' opinion has changed in response to this, with many expressing expectations that the government will investigate the facts and fulfill their legal obligations.

Through the semantic relationship network diagram, it can be seen that Weibo users focus their attention on the current situation of food safety, dissatisfaction with the handling results of relevant departments, and other aspects of this incident. The specific focus can be determined through thematic clustering.

Perplexity is a measure of how uncertain documents are about each issue; lower values suggest better model performance. Perplexity was computed for each of the 15 topics in this study, and a topic perplexity curve figure was created (Fig 6). The figure indicates a progressive decrease in bewilderment with an increase in the number of topics. However, the change in confusion becomes less substantial when the number of topics is set to six. In order to guarantee the correctness of the analysis results, we decided to limit the number of topics to six using the Occam's razor approach.

Assuming 0.1 and 0.01 for α and β, respectively, Fig 7 displays the partial LDA topic clustering results, with each bubble denoting a subject. The graphic shows that the six subject bubbles are pretty far off from one another and do not have a lot of overlapping areas, which suggests that the LDA model did a good job in dividing the topics within the text. Table 3 displays the top 10 topic terms for each topic. Each topic is summarized below.

Topic 1: From the "bureau of management", "survey" and "information bulletin" and other subject terms and with specific text can be seen in the school was exposed to food safety incidents, the relevant personnel immediately on the incident and the person involved in the investigation. After the exposure of the food safety incident in the school, the relevant personnel immediately launched an investigation into the incident and the person involved, and gave the result of the investigation that the foreign body eaten by the student was a "duck neck".

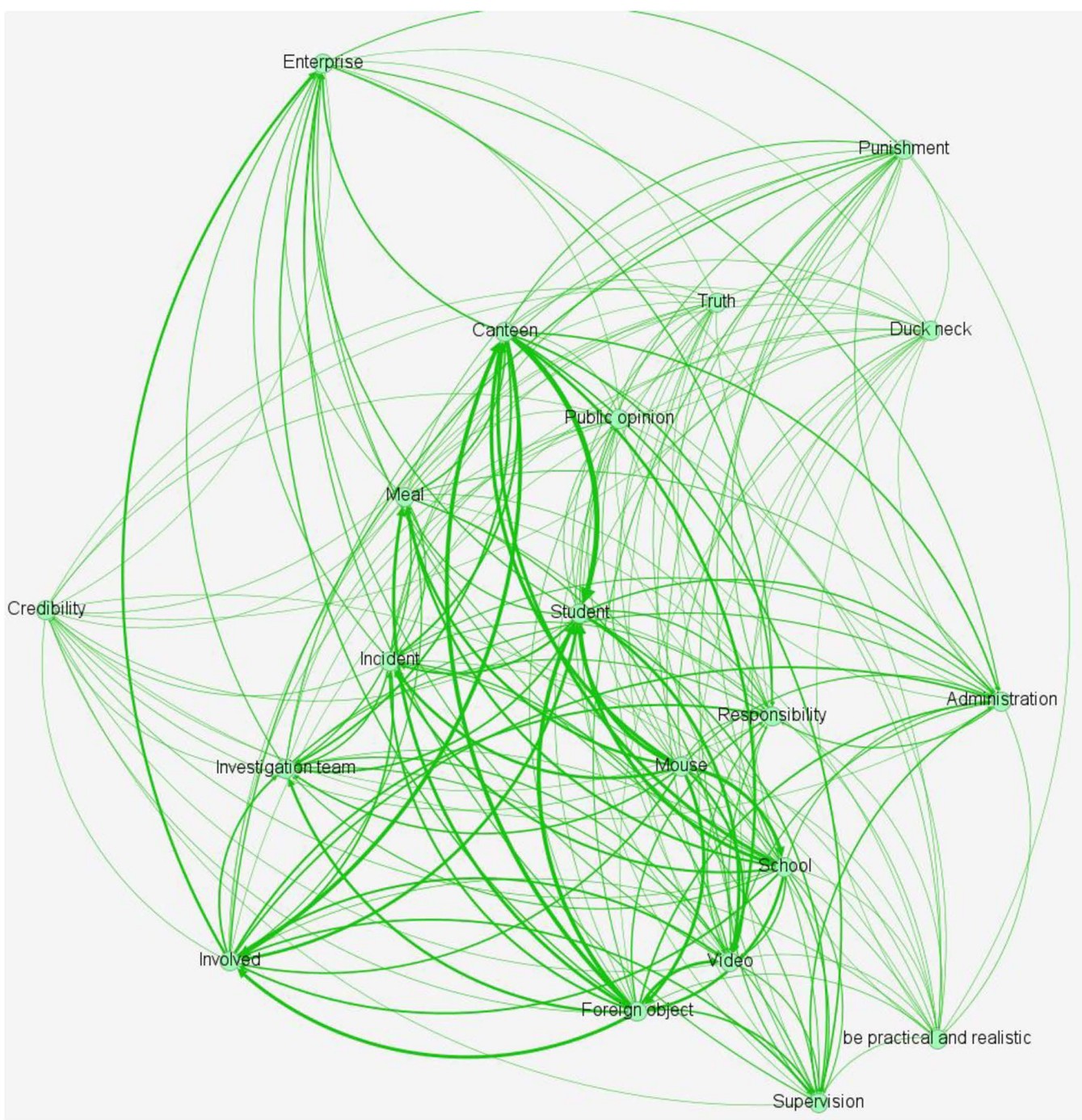

**Fig 5. Semantic Relationship Network Figure.**

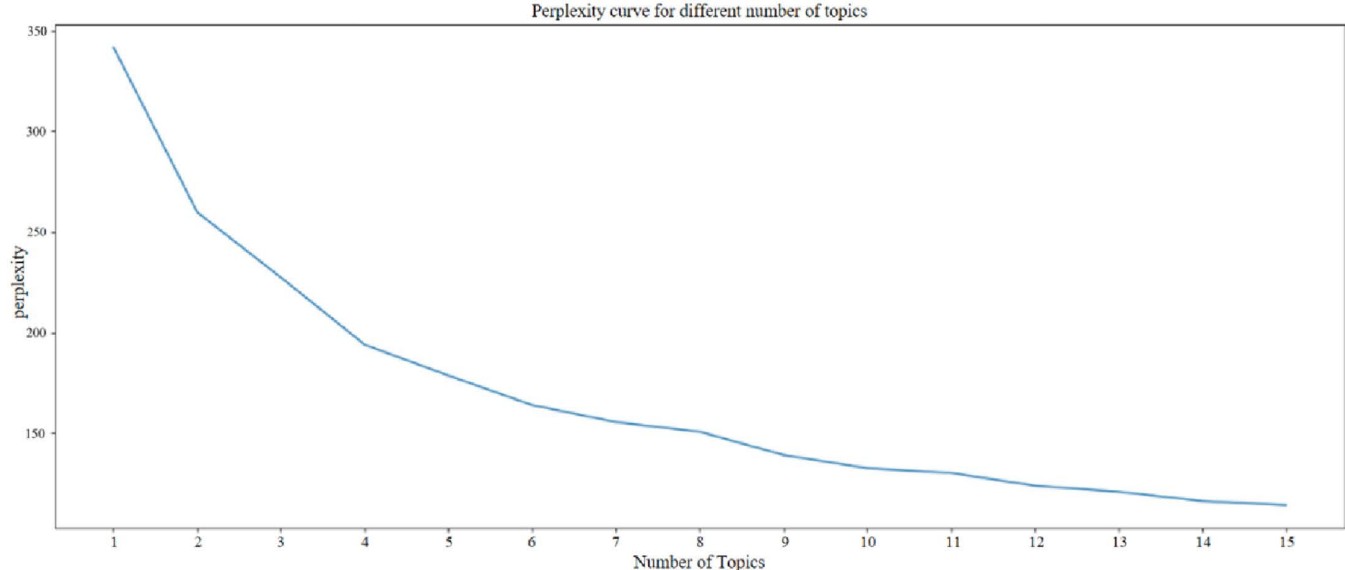

**Fig 6. Topic Perplexity Figure.**

Topic 2: From the topics of "ancient", "call a deer a horse"(In Chinese culture, 'call a deer a horse' refers to deliberately distorting the truth to mislead others, similar to gaslighting.), and "blindness" combined with specific texts, it can be seen that Weibo users are angry about the exposed campus food safety incident and the subsequent handling of related personnel, expressing their dissatisfaction with the deception of the public by related personnel through the allusion of "pointing deer as a horse".

Topic 3: From the thematic Words "confirmation", "finally" and "mouse's head", and in conjunction with the specific text, it can be seen that through repeated investigations by the Market Supervision Bureau and the identification by authoritative animal experts, it was finally determined that the foreign substance eaten by the student in the cafeteria was a mouse head.

Topic 4: From the "fall", "sales counter" and "Zhou Hei Ya" these words and combined with the specific text can be seen due to the food safety incident on the hot search, "Zhou Hei Ya" and other products have also been implicated, not only has the normal business of the relevant stores been affected, but the company's share price has also begun to fall continuously. "Zhou Hei Ya" and other products have also been implicated, not only the normal business of the relevant stores have been affected, and even the company's share price has also begun to fall continuously.

Topic 5: From the topic words "bug", "picture" and "complaint" and the specific text, it can be seen that not long after the school was exposed to the student's report of eating mouse head, some student claimed to have eaten bugs in the cafeteria and complained about it. The school was exposed to a student who ate a mouse's head, and another student complained about eating bugs in the cafeteria.

Topic 6: From the key words "food safety", "credibility" and "public opinion" and the specific text, it can be seen that the outbreak of the food safety incident and the recent food safety situation have created a great public opinion on the Internet, which has a certain impact on the government's credibility. The outbreak of the food safety incident and the recent food safety situation have created a huge public opinion on the Internet, which has had a certain impact on the credibility of the government.

Based on the results of text mining and combined with specific content, it can be seen that after the outbreak of food safety incidents, the public immediately paid attention to this matter and engaged in fierce online debates, leading to the

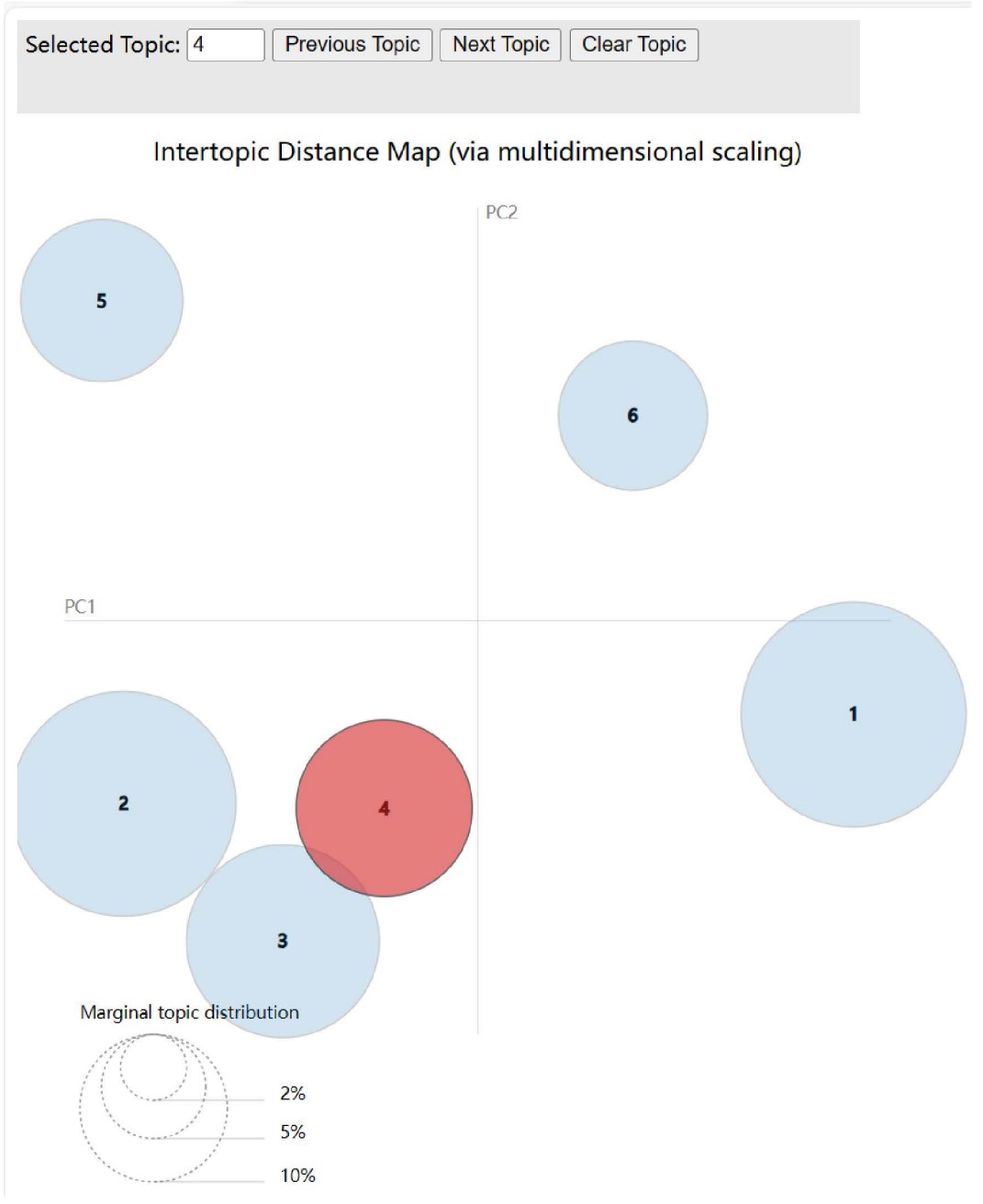

**Fig 7. LDAvis Visualization Figure.**

emergence of network public opinion. Corresponding inquiries were also started by schools and pertinent departments. Following staff confirmation and testing by inspection agencies, a notice was published claiming that the foreign objects discovered by kids in the cafeteria were "duck necks" The people, however, rejected this explanation and persisted in vehemently debating it online, which fueled more fervor in the public opinion there. It was eventually established that the foreign body was a mouse head following an inquiry by the Jiangxi Province Joint inquiry Team and identification by

Table 3. Distribution of Topic words.

| Topic 1 | bureau of management | supervision | scene | involved | survey | foreign body | video | publish | party involved | information bulletin |
|---|---|---|---|---|---|---|---|---|---|---|
| Topic 2 | tooth | inspection | beard | probability | ancient | blindness | incisor | call a deer a horse | zhao gao | modern version |
| Topic 3 | meals | confirmation | foreign body | finally | response | convince | law enforcement | municipal supervision bureau | mouse's head | Rashomon |
| Topic 4 | Juewei | stock | fall | victim | listed company | shareholder | sales counter | Zhou Hei Ya | discrepancy | market open |
| Topic 5 | dish | complaint | moth | school cafeteria | bug | devastated | picture | steel wire | feedback | disgusting |
| Topic 6 | food safety | confusing | rumor | credibility | public opinion | government | tacitus trap | food | no trivial matters | investigation team |

animal specialists. Soon later, pupils reported that they had discovered insects in the food in the cafeteria. The public discourse about the food safety event on campus and the state of food safety in general peaked at this period. Certain food companies associated with "duck necks" including "Zhou Hei Ya" were also involved, casting doubt on the trustworthiness of the relevant departments. This resulted in a persistent drop in stock values as well as certain detrimental effects on economy.

This abrupt public health emergency shows that, following news from multiple media sources, the public's primary concerns are the facts of the incident and the ability of the appropriate departments to act promptly and effectively. The direction of network public opinion can be effectively guided, and the harm caused by it can be minimized, provided the relevant departments can act quickly to recognize public demands, investigate the outbreak of network public opinion, and respond to it.

In the previous section, we quantitatively extracted and analyzed the topics of public concern during the outbreak of network public opinion on food safety incidents, and initially revealed the hotspots of public opinion and the characteristics of topic distribution. However, although quantitative analysis can identify the core topics of public concern, it is difficult to deeply reveal the formation mechanism and dynamic evolution of network public opinion. Therefore, in order to understand the logic of network public opinion generation in a more systematic way, this chapter will adopt the qualitative research method of grounded theory to construct a model of network public opinion generation through layer-by-layer coding and analysis. The influencing factors and interactions of public opinion generation are discussed in depth.

Due to the large amount of data in this experiment, the traditional grounded theory, which completely relies on manual article-by-article coding, has limitations in efficiency and operability, and is difficult to be directly applied to the research scenario of this paper. In addition, in the text mining process in the previous section, we have extracted the topic keywords under six topics through the topic model and screened out the high-weight words that can represent the core information of the text by combining them with the TF-IDF values. The tag words composed of these keywords and high TF-IDF value words not only cover the main content of the text, but also reflect the key features of the data. Therefore, in this paper, these tag words are used as the basis instead of the original data as the main reference basis for coding. This method not only significantly reduces the workload of manual coding and improves the efficiency of the study, but also maximizes the avoidance of subjective bias and ensures the objectivity and neutrality of the coding process. At the same time, this method also makes the coding results organically connected with the text mining section above, providing more reliable data support for the construction of the subsequent online opinion generation model.

Open coding begins with the basic concept definition, which serves as the foundation for the creation of initial categories. It entails developing the text's basic conceptual framework. Label words are directly employed as the coding findings

**Table 4. Initial Conceptualization Results.**

| Tag Words | Tag Words | Tag Words | Tag Words | Tag Words |
|---|---|---|---|---|
| Cafeteria | The Paper | Seek Truth from Facts | Absurd | Consumer Rights |
| Mouse | Testing | Report | Rights Protection | Law Enforcement Officials |
| Student | Conclusion | Trust | Distort the Truth | Call a Deer a Horse |
| Foreign Body | Credibility | Hair | Vicious | Rodent |
| Meal | Journalist | Rectify | Disgusting | Fermentation |
| School | Stock | Refute a Rumor | Oriental Web | Legal Representative |
| Video | Education Department | Outrageous | Law | Not be Convinced |
| Investigation Team | Weibo | Apologize | Sadness | Deception |
| Confirmation | Unimaginable | Illegal | Zhao Gao | Hot Search |
| Supervision | Revoke | Food Hygiene | Rashomon | Dissemination |
| Response | License | Shareholder | Government | Modern Version |
| Bureau of Management | Heavier than Mount Tai | Fall | Tacitus Trap | Fool |
| Responsibility | Intervene | Rumor Mongering | Devastated | Calm Down |
| Duck Neck | Person In Charge | China News Service | Blindness | Contractor |
| Identify Mouse as Duck | Exposure | Lie | Complaint | Internet |
| Food Safety | Identification | Catering Company | Yangtze Evening News | The Truth Comes to Light |
| Penalty | Safety Law | Obscure the Truth | Strongly Dissatisfied | Joke |
| Person Involved | Believe | Rule of Law | Expert | Strict |

of the study's first notions since they are extremely thorough and succinct. Table 4 displays the selection of 90 label terms for this paper based on topic words and words with high TF-IDF values.

Initial category definition is the second step of open coding. In the first step of open coding, we screened 90 tag words as the coding basis for the initial concepts based on the topic words extracted from the LDA topic model and the words with high TF-IDF values. These tag words were selected based on their high frequency occurrence in the text and their core reflection of the opinion events. For example, the word "cafeteria" appeared frequently in the text and was closely related to the location of the incident, so it was categorized as "location of the incident"; and the word "mouse" was directly associated with the incident The word "mouse", on the other hand, is directly related to the central point of contention of the incident, i.e., whether the foreign body eaten by the student was a mouse head or not, and is therefore categorized as "truth of the incident". In this way, we ensure that the categorization of each tag word is closely related to its semantic role in the text and to the core elements of the opinion event. By editing different textual contents into independent nodes and integrating nodes of the same category into a category according to relevant principles, the category is finally named scientifically. According to the 90 tag words selected in this paper, combined with the LDA thematic clustering results and the specific content of the text, and with reference to the coding results of previous researchers and discussions with other graduate students, 20 initial concepts were finally summarized, and the results of the division of the initial concepts are shown in Table 5.

The first set of classification results has been divided into twenty categories: location of the incident; subject of the incident; nature of the incident; truth of the incident; government organization; investigation; government notice; means of handling; government credibility; social impact; people's sentiment; people's expectation; people's feedback; people's right; acting in accordance with the law; responsibility; seek truth from facts; be honest and upfront; dissemination and media. Axial coding of these initial categories is the next phase.

Principal axis coding is the process of providing a higher level of generalization of the initial categories formed by the previous open coding, forming the main categories, narrowing down and refining the core concepts. This process aims

**Table 5. Initial Categorization Results.**

| Tag Words | Initial concepts | Tag Words | Initial concepts | Tag Words | Initial concepts | Tag Words | Initial concepts | Tag Words | Initial concepts |
|---|---|---|---|---|---|---|---|---|---|
| Cafeteria | Location of the Incident | The Paper | Media | Seek Truth from Facts | Seek Truth from Facts | Absurd | Nature of Incident | Consumer Rights | People's Expectation |
| Mouse | Truth of the Incident | Testing | Investigation | Report | Dissemination | Rights Protection | People's Feedback | Law Enforcement Officials | Incident Investigation |
| Student | Subject of the Incident | Conclusion | Truth of the Incident | Trust | Be Honest and Upfront | Distort the Truth | Nature of Incident | Call a Deer a Horse | People's Sentiment |
| Foreign Body | Subject of the Incident | Credibility | Government Credibility | Hair | Subject of the Incident | Vicious | Nature of Incident | Rodent | Truth of the Incident |
| Meal | Subject of the Incident | Journalist | Dissemination | Rectify | Means of Handling | Disgusting | People's Sentiment | Fermentation | Dissemination |
| School | Location of the Incident | Stock | Social Impact | Refute a Rumor | Government Notice | Oriental Web | Media | Legal Representative | Subject of the Incident |
| Video | Investigation | Education Department | Government Organization | Outrageous | Nature of Incident | Law | People's Right | Not be Convinced | People's Feedback |
| Investigation Team | Investigation | Weibo | Media | Apologize | Government Notice | Sadness | Nature of the Incident | Deception | Government Credibility |
| Confirmation | Investigation | Unimaginable | Nature of Incident | Illegal | Nature of Incident | Zhao Gao | People's Sentiment | Hot Search | Dissemination |
| Supervision | Investigation | Revoke | Means of Handling | Food Hygiene | People's Expectation | Rashomon | Nature of Incident | Dissemination | Dissemination |
| Response | Government Notice | License | Means of handling | Shareholder | Social Impact | Government | Government Organization | Modern Version | People's Sentiment |
| Bureau of Management | Government Organization | Heavier than Mount Tai | People's Expectation | Fall | Social Impact | Tacitus Trap | Government Credibility | Fool | People's Sentiment |
| Responsibility | Responsibility | Intervene | Investigation | Rumor Mongering | Dissemination | Devastated | People's Sentiment | Calm Down | Means of Handling |
| Duck Neck | Subject of the Incident | Person in Charge | Investigation | China News Service | Media | Blindness | People's Sentiment | Contractor | Subject of the Incident |
| Identify Rat as Duck | People's Sentiment | Exposure | Dissemination | Lie | Government Credibility | Complaint | People's Right | Internet | Media |
| Food Safety | People's Expectation | Identification | Incident Investigation | Catering Company | Subject of the Incident | Yangtze Evening News | Media | The Truth Comes to Light | Truth of the Incident |
| Penalty | Means of Handling | Safety Law | People's Right | Obscure the Truth | Means of Handling | Strongly Dissatisfied | People's Sentiment | Joke | Nature of the Incident |
| Person Involved | Subject of the Incident | Believe | Be Honest and Upfront | Rule of Law | Acting in Accordance with the Law | Expert | Investigation | Strict | People's Expectation |


to identify a few key areas that are most representative and influential from the many initial categories, providing a solid foundation for subsequent theoretical model construction. Through further analysis and synthesis, we summarized and enhanced the initial categories and combined them with the coding results of previous researchers to finally obtain six main categories. These master categories not only cover the key elements of the event, but also reveal the roles and interaction styles of different subjects in the event. The specific results are shown in Table 6.

In this paper, six main categories, namely, people's responses, value orientations, news media, public opinion effects, national subjects, and public food safety incident, are obtained through principal axis coding to provide the basis for the

**Table 6. Extracted Primary Categories.**

| main category | Initial scope | main category | Initial scope |
|---|---|---|---|
| Food Safety Incident | Location of the Incident | People's Responses | People's Sentiment |
| | Truth of the Incident | | People's Expectation |
| | Subject of the Incident | | People's Feedback |
| | Nature of Incident | | People's Right |
| National Subjects | Government Organization | Value Orientations | Acting in Accordance with the Law |
| | Investigation | | Responsibility |
| | Government Notice | | Seek Truth from Facts |
| | Means of Handling | | Be Honest and Upfront |
| Public Opinion Effects | Government Credibility | News Media | Dissemination |
| | Social Impact | | Media |

construction of the network public opinion generation model, and the specific interpretations of the six main categories are as follows:

(1) Food safety incident: as the core background of network public opinion generation, it covers the location, truth, subject and its nature of the event. Together, these factors constitute the basic framework of the event and influence the public's focus and direction of discussion.

(2) People's responses: It includes the public's sentiment, expectations, feedback, and rights defense behavior. This category reflects the public's attitudes and actions toward the event, and is an important part of network public opinion, directly affecting the development and direction of public opinion.

(3) National subjects: It involves the roles of governmental organizations, such as the investigation of the incident, the release of notices, and the means of handling. The behavior and policies of state subjects are directly related to the public's trust and the efficiency of the incident's resolution, and they are the key force in guiding public opinion.

(4) Value orientations: It covers the principles of acting in accordance with the law, responsibility, seeking truth from facts, and being frank. These values not only influence the behavior of the government and the public, but also shape the society's perception and evaluation standards of the incident.

(5) Public opinion effects: focuses on government credibility and social impact, and assesses the impact of the incident on social stability and public confidence. Public opinion effects reflect the social repercussions and long-term consequences triggered by the incident.

(6) News media: As the main channel for information dissemination, it includes event communication and communication media. The media play the role of a bridge and amplifier in the generation and development of public opinion, influencing public perceptions and attitudes.

The process of selective encoding in grounded theory is also the process of constructing a model. By analyzing the results of extracting main categories at a deeper level, all main categories are organically connected through a "storyline" to form a theoretical model, and the logical relationships between nodes in the model are reasonably explained. Based on the six main categories extracted earlier, this article constructs a network public opinion generation model for food safety incidents, as shown in Fig 8. (It is also worth noting that in the open coding stage, we screened 90 labeled words as the basis for coding the initial concepts based on the topic words extracted from the LDA topic model and the words with high TF-IDF values. Subsequently, by communicating with our classmates and referring to the coding results of previous researchers, we integrated these initial concepts into 20 initial categories. In the spindle coding stage, we further

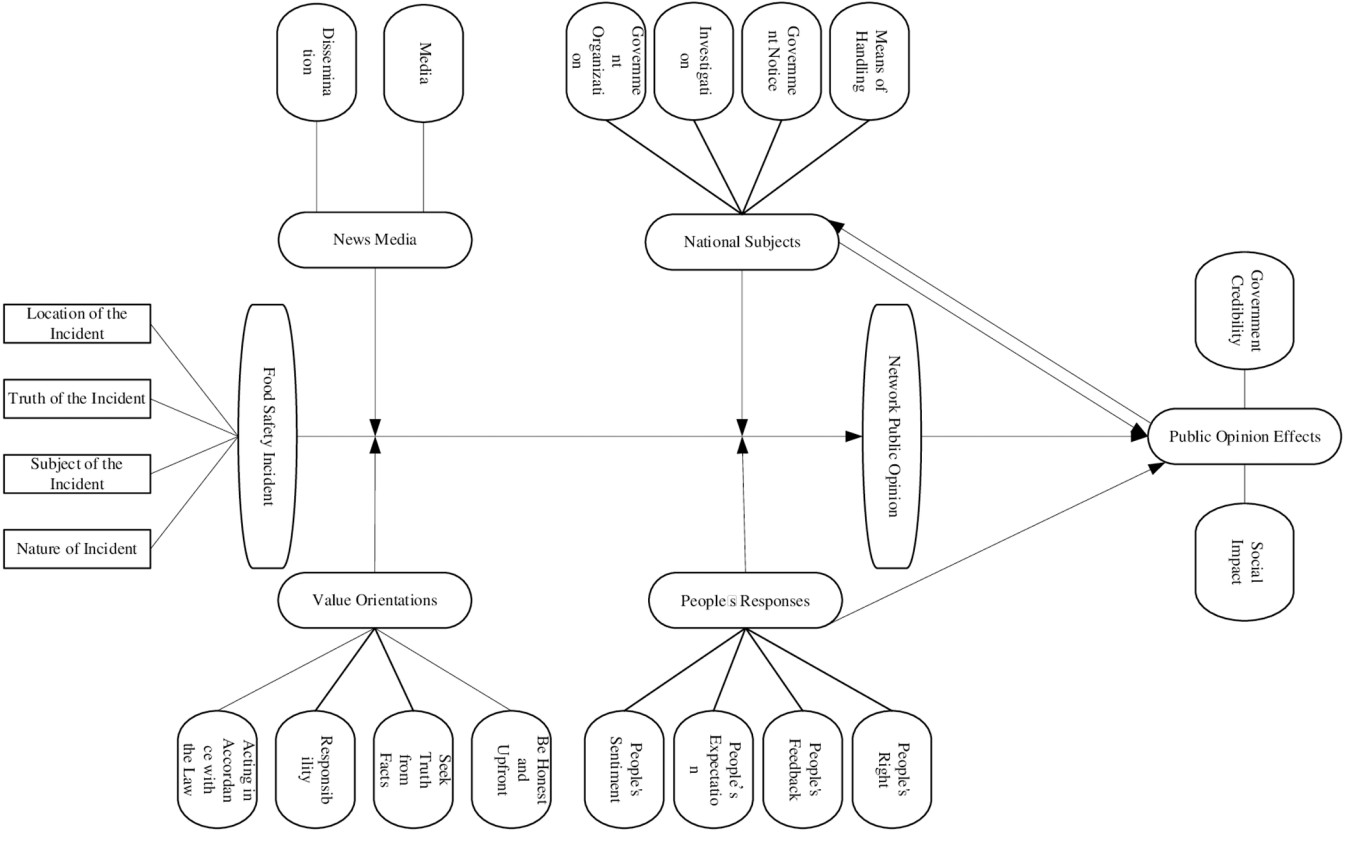

**Fig 8. Network Public Opinion Generation Model for Food Safety Incident.**

generalized these initial categories into 6 main categories and constructed an online opinion generation model. This process is similar to the "theory memoing" in traditional grounded theory, which helps us to gradually distill the core theory).

According to Fig 8, it can be seen that after the outbreak of a food safety incident, influenced by value orientations, the public will respond to the incident in various ways, and network public opinion will begin to erupt, resulting in various negative impacts. The relevant government departments respond to the outside world by investigating the truth of the incident and taking various measures to guide the direction of network public opinion and reduce the negative impact it brings. The public evaluates the responses made by relevant departments to generate positive or negative feedback on network public opinion, and expects the government to respond again. This process is repeated until the public opinion gradually subsides, which is the basic process of generating network public opinion after the outbreak of food safety incidents. In addition, through saturation testing, no other new categories or concepts were found, indicating that the model theory has reached saturation.

The preceding text discusses the generation model of network public opinion on food safety incidents by integrating it with specific practices and analyzes the process of generating such public opinion. Based on this, the following section will analyze the various participating entities in the process of public opinion formation and provide relevant recommendations.

This study found that the main subjects involved in the formation of network public opinion mainly include public opinion subjects, public opinion intermediaries, public opinion objects and government forces. These subjects are closely related to the main categories of grounded theory (e.g., "people's responses," "news media," and "national subjects"), and together they contribute to the generation of network public opinion.

The first is the subject of public opinion. The main index of public opinion is the large number of online users, whose focus plays an important role in generating public opinion. Based on the main category of "public's responses" in grounded theory, the focus of the public opinion subject is the core driving factor for the generation of public opinion. With the development of new media, the role of opinion leaders in public opinion dissemination is becoming increasingly prominent. For example, in this food safety incident, some major influencers spread rumors of "duck neck" before the truth of the incident was clarified, which had a negative impact on the development of public opinion. Therefore, guiding the focus of public opinion subjects through science popularization and online literacy education is an important means to alleviate public opinion.

The second is the object of public opinion. The object of public opinion refers to the event itself that triggers public opinion, which is usually closely related to public safety and interests. In this incident, the food safety of a university in Jiangxi Province became the focus of public opinion. The nature of the object of public opinion (e.g., the seriousness and transparency of the incident) directly affects the development of public opinion, which is closely related to the main category of "value orientations" in the grounded theory. The public's pursuit of truth and expectation of fairness and justice are important driving forces for the generation of public opinion.

The third is public opinion intermediaries. Public opinion intermediaries mainly refer to the information platforms that public opinion dissemination relies on, such as Sina Weibo, Jitterbug, WeChat, and so on. Based on the main category of "news media" in the grounded theory, opinion intermediaries play a key role in information dissemination and opinion guidance. For example, in this incident, the school and some members of the media tried to cover up the truth by fabricating rumors, which, on the contrary, aggravated the public's dissatisfaction. It was only with the disinformation of the mainstream media and the intervention of the investigation team that public opinion was alleviated. Therefore, strengthening the regulation and guidance of public opinion intermediaries is an important part of public opinion governance.

The fourth is the power of the government. The role of the government in public opinion events is highly compatible with the main category of "national subjects" in the grounded theory. Transparent response and credibility of the government is the key to alleviate public opinion. In this incident, the initial responses of the Nanchang City Market Supervision Administration and the school failed to quell the public's skepticism, and it was only after Jiangxi Province set up a joint investigation team and publicized the truth that public opinion was gradually brought under control. This shows that timely response and effective action by the government are crucial to public opinion management.

Internationally, the governance of public opinion has become an important issue of concern to governments and society, and the United Nations report on cybergovernance calls on all countries to establish a sound system of comprehensive cybergovernance to meet the challenges of disinformation and cyberopinion. This shows that improving the system of public opinion monitoring, improving the mechanism for guiding major public opinions and emergencies, and creating a clear cyberspace are common goals of global public opinion governance. Based on the network public opinion generation model constructed in this study, we identified four key main categories: national subjects, people's responses, value orientations and news media. The following recommendations aim to propose specific public opinion management strategies for these main categories in order to more effectively minimize the harm caused by network public opinion.

Based on the two main categories of news media and national subjects, we suggest establishing a cross-platform public opinion monitoring and collaborative response mechanism. Information platforms (e.g., Weibo, Jitterbug, Twitter, Facebook) are the core channels for opinion generation and dissemination, and the government should cooperate with major platforms to establish a real-time opinion monitoring system to detect and respond to false information and rumors in a timely manner. At the same time, the government should coordinate information sharing and collaborative response among different platforms to ensure consistency and effectiveness of public opinion response. This strategy applies not only to China but also to other countries, especially in the globalized information environment, where cross-platform collaboration is the key to curbing disinformation and guiding the direction of public opinion.

Based on the two main categories of people's responses and value orientations, we suggest promoting public participation in opinion management. Netizens' emotions and value orientations are important drivers of public opinion generation. Therefore, the government should encourage public participation in public opinion governance, for example, by setting up a public reporting mechanism, launching education campaigns on Internet literacy, and organizing public discussions on food safety incidents. Such participatory governance not only improves the public's ability to recognize food safety incidents, but also enhances the public's understanding of and support for government decisions. This strategy has global applicability, especially in democratic societies where public participation is an important way to enhance the effectiveness of public opinion governance.

Based on the two main categories of national subjects and news media, we suggest building a data-based public opinion early warning and decision support system. The government can use big data and artificial intelligence technology to analyze the dynamics of public opinion on social media platforms in real time, predict the development trend of public opinion, and provide decision makers with a scientific basis. For example, by analyzing netizens' emotional tendencies and information dissemination paths, the government can identify potential public opinion risks in advance and take targeted countermeasures. This strategy is not only applicable to China, but also provides a reference for other countries. Especially when dealing with public health emergencies, a data-driven decision support system can significantly improve the government's response efficiency and credibility.

Based on the two main categories of news media and value orientations, we propose to promote media responsibility and self-regulation mechanisms. The media plays an important role in the generation and dissemination of public opinion, and therefore the media's sense of responsibility and self-regulation should be strengthened. For example, the government can encourage media organizations to develop industry norms to ensure the objectivity and accuracy of reporting, as well as set up a third-party monitoring organization to assess and hold media accountable for their reporting behavior. This strategy applies not only to China but also to other countries, especially in the age of information explosion, where media responsibility and self-discipline are key to maintaining public trust and social stability.

## Conclusion

This article is based on a sudden food safety incident in a university in Jiangxi Province, combining text mining and grounded theory to apply to the study of network public opinion. This study first uses text mining methods to identify the main concerns of the public during public opinion periods. Based on this, topic words and keywords with high TF-IDF values are extracted as the basis for grounded theoretical coding. Finally, a model for generating network public opinion on food safety incidents is constructed. By analyzing the model, it was found that the four major participants in the formation of network public opinion are the public opinion subject, public opinion intermediary, public opinion object, and government power. Targeted suggestions were proposed from three perspectives: public opinion subject, public opinion intermediary, and government power. There are also some shortcomings in the research: The selected food safety incidents were collected from campus food safety incidents, and the recommendations and conclusions given may not be comprehensive enough, and there is a need to select more and more comprehensive food safety incidents to be studied in the future. In addition, due to the qualitative nature of grounded theory, which inherently involves a degree of subjectivity, we used a hybrid approach to increase the rigor and objectivity of the coding process. Specifically, we combined LDA topic modeling with grounded theory and used the extracted topic keywords as the initial data for coding. This approach not only reduces the potential bias of purely manual coding, but also provides a data-driven basis for identifying key topics. To further ensure the reliability of the coding, we implemented a data triangulation procedure. However, we acknowledge that future research could explore more systematic or software-assisted coding methods to further improve the reproducibility and objectivity of the findings. Moreover, this study is an analysis of the generation of network public opinion, and the research focuses on topic extraction and the core elements of public opinion generation, so it focuses more on static categorization and thematic analysis rather than dynamic processual, and although it is able to quickly identify the core topics and key

elements of an event, its limitation is that it cannot adequately capture the dynamic evolution of public opinion, and future research can further integrate processual analysis to more comprehensively understand the process of generating and evolving network public opinion.

## Author contributions

**Conceptualization:** Zheyu LIN, Wei YANG.

**Data curation:** Zheyu LIN, Wei YANG.

**Formal analysis:** Zheyu LIN, Wei YANG.

**Funding acquisition:** Jiayin PEI.

**Investigation:** Zheyu LIN.

**Methodology:** Zheyu LIN, Wei YANG.

**Project administration:** Zheyu LIN.

**Resources:** Zheyu LIN, Wei YANG.

**Software:** Zheyu LIN.

**Visualization:** Zheyu LIN, Wei YANG.

**Writing – original draft:** Zheyu LIN, Wei YANG.

**Writing – review & editing:** Jiayin PEI, Wei YANG.

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
