## [Decision Letter · Decision Letter 0]

Dear Dr. LIN,

Thank you for submitting your manuscript to PLOS ONE. After careful consideration, we feel that it has merit but does not fully meet PLOS ONE’s publication criteria as it currently stands. Therefore, we invite you to submit a revised version of the manuscript that addresses the points raised during the review process.

We look forward to receiving your revised manuscript.

Kind regards,

Tinggui Chen

Academic Editor

PLOS ONE

Journal Requirements:

2. In your Methods section, please include additional information about your dataset and ensure that you have included a statement specifying whether the collection and analysis method complied with the terms and conditions for the source of the data.

3. Please note that PLOS ONE has spec6ific guidelines on code sharing for submissions in which author-generated code underpins the findings in the manuscript. In these cases, all author-generated code must be made available without restrictions upon publication of the work. Please review our guidelines at https://journals.plos.org/plosone/s/materials-and-software-sharing#loc-sharing-code and ensure that your code is shared in a way that follows best practice and facilitates reproducibility and reuse.

5. Please provide a complete Data Availability Statement in the submission form, ensuring you include all necessary access information or a reason for why you are unable to make your data freely accessible. If your research concerns only data provided within your submission, please write "All data are in the manuscript and/or supporting information files" as your Data Availability Statement.

Additional Editor Comments :

I have completed my evaluation of your manuscript. The reviewers recommend reconsideration of your manuscript following major revision. I invite you to resubmit your manuscript after addressing the comments below.

Reviewers' comments:

Reviewer's Responses to Questions

**Comments to the Author**

1. Is the manuscript technically sound, and do the data support the conclusions?

Reviewer #1: Yes

Reviewer #2: Yes

Reviewer #3: Partly

Reviewer #4: Partly

Reviewer #5: Partly

Reviewer #6: Partly

2. Has the statistical analysis been performed appropriately and rigorously?

Reviewer #1: Yes

Reviewer #2: Yes

Reviewer #3: I Don't Know

Reviewer #4: I Don't Know

Reviewer #5: I Don't Know

Reviewer #6: Yes

3. Have the authors made all data underlying the findings in their manuscript fully available?

Reviewer #1: No

Reviewer #2: Yes

Reviewer #3: No

Reviewer #4: No

Reviewer #5: Yes

Reviewer #6: Yes

4. Is the manuscript presented in an intelligible fashion and written in standard English?

Reviewer #1: Yes

Reviewer #2: Yes

Reviewer #3: Yes

Reviewer #4: No

Reviewer #5: Yes

Reviewer #6: Yes

Reviewer #1: I have reviewed your manuscript "Research on the Generation of Food Safety Network Public Opinion - Taking the Chinese Rat Head Duck Neck Incident as an Example." The paper addresses an important topic in food safety and public opinion analysis, and while revisions are needed, it shows promise in contributing to our understanding of how public opinion forms around food safety incidents.

The manuscript examines a relevant social issue, and your comprehensive data collection approach using Weibo data provides a solid foundation for analysis. The case study selection is compelling and timely. However, several aspects of the paper require attention.

The literature review, while comprehensive in its coverage, currently reads more like a catalog of previous work rather than a critical synthesis. A more analytical approach to reviewing previous research would better highlight the gaps in current knowledge and more clearly position your contribution to the field. Consider restructuring this section to draw out key themes and debates in the literature rather than presenting a chronological list of studies.

Regarding methodology, while your approach is systematic, the technical exposition could be more focused. The current detailed step-by-step description of grounded theory and topic selection processes, while thorough, tends to overshadow the paper's main contributions. Consider condensing these methodological details and focusing more on your analytical insights. Additionally, while perplexity is used as a metric for topic modeling evaluation, the paper would benefit from a clear justification of why this metric was chosen over other available options such as probabilistic coherence or topic coherence measures. This justification would strengthen the methodological decisions made in the study.

The data analysis shows interesting findings, particularly in the six identified themes, though their relationship to the broader theoretical framework of public opinion formation could be further developed. The transition from LDA results to grounded theory coding needs more explanation to show how analytical depth was ensured.

The implications of your research for food safety governance are promising but could be more fully developed. More specific insights drawn from your analysis would enhance the paper's practical value for stakeholders managing food safety public opinion issues. The writing would benefit from some editing for clarity and flow, and the theoretical framework could be more clearly articulated.

I recommend moderate revisions focusing on:

- Restructuring the literature review to provide a more analytical synthesis

- Streamlining the methodological presentation

- Justifying the choice of validation metrics

- Deepening the analytical discussion

- Enhancing the practical implications

The paper addresses an important topic and has the potential to contribute meaningful insights about public opinion formation in food safety incidents. I look forward to reviewing a revised version that builds on its current strengths while addressing these suggestions for improvement.

Reviewer #2: You can improve your literature review writing by reading journals and making notes of what made their lit reviews easy or hard to understand.

One misconception that exists is that the harder it is to understand a lot review the better it must be! We often think we're not smart enough to understand when REALLY it's the fault of the writers who do not write clearly. Use clear, simple sentences and only all detail or jargon when it adds to the meaning and nuance of your sentence.

Reviewer #3: Well written, in clear and simple English. Initially well-structured but could be more explicit in how the study was conducted. It would benefit from revising to make it clearer and also by using the COREQ guidelines to interrogate how reflexivity can aid such research.

Interesting use of data-mining together with well founded approaches to algorithm data analysis (LDA and TF-IDF). I have a few comments as I think the paper has relevance to a wide audience who will have different levels of familiarity with the methods of use. You will thus potentially need to spell out exactly what you have done at each point, and how. This will make the paper very useful to many other researchers. Specific points are:

- I think you need to discuss the use (without permission) of social media posts in research. There are now some ethics concerns and these may be reasonable but you need to be transparent about this.

- I think you need to justify removal of emojis especially, as in social media posts these can convey a significant amount of nuance which is a key critique of using algorithmic data analysis techniques.

- I am not clear what methods (if any) of the study were manually undertaken e.g. in 4.1. 'by combining synonyms for the separated term...' how did the researchers do this. It might help to write such parts of the paper in a more active tense, or even use the first person, to highlight how different aspects of the study were conducted. This is important as I am not sure the interpretation of co-occurring words (and what they signify) on page 23 is the only one. I am not clear how you defined clusters of co-occuring words; I can see you have used a form of SNA (at least to create a graph). From the text I am assuming your interpretation of co-occurring is based on a visual analysis of the graph or did you analyse this using a traditional clustering method (e.g. louvain on igraph etc). I think it would be useful to provide more detail on this.

- I am not entirely clear how you defined the initial 15 topics; I believe this is informed by the researcher. I can see you used perplexity scores to optimise the number of topics (eventually to six). Did you use any other scores e.g. coherence etc. If not, justify this.

- I can understand the application of GT to your data; I think some of the nuance of GT is lost by the approach you have taken. Ideally GT focusses on processes ('gerunds') which isn't the case. A future analysis could perhaps do this but I think, in your case, it is worth discussing the benefits and drawbacks of the approach you have taken and that by doing so it perhaps neglects the focus on process.

- I think rather than aim to be impartial/ neutral it might be better to understand the reflective biases within any type of research (including this) - I have given some suggestions on this at the end of this review.

- again in your GT analysis you need to be clear on how you moved from e.g. spatial to axial coding;I think you need to be explicit e.g. I/ Researcher X developed the following codes by...

- similarly it would be useful to explain the storyline. Again in conventional GT researchers use theoretical memoing; it sounds like this fulfils a similar role so I think you need to explain it and potentially draw the parallel.

I think it would be critical to try and assess this paper using the COREQ framework; while designed for more conventional qualitative research doing this would provide opportunities to

a) systematically justify and highlight the strengths of both computer-aided/ algorithmic analyses and human input

b) allow a discussion of reflexivity and what this means in this context

c) broaden the relevance of this paper methodologically, as the opportunities for machine-learning / NLP aided analyses and qualitative syntheses exponentially increase in the coming years.

Reviewer #4: As a journal reviewer, I offer the following suggestions for your paper:

1. Missing All Figures:

Your manuscript is missing crucial figures. These figures are essential for understanding and evaluating your research findings. Please include these figures to support your analysis and conclusions.

2. Chapter 2 Lacks Original Insights; Suggest Refining the Related Work Section:

Chapter 2 primarily presents a mechanical repetition of existing literature without your own perspectives or in-depth analysis. I incorporate critical discussions of related research to showcase your unique insights and understanding of the field.

Additionally, consider further subdividing the related work section. For example, categorize frameworks based on different research focuses within the academic field (including empirical studies). This will help readers better comprehend the context of existing research. Currently, the section is overly lengthy and somewhat redundant; appropriate condensation and refinement will enhance the paper's readability.

3. Streamline the Research Methodology in Chapter 3; Emphasize Research Process and Data Handling:

Chapter 3 provides excessively detailed explanations of the origins, internal logic, and formulas of the research methods. It is advisable to simplify these theoretical descriptions and focus on the specific research procedures and data processing methods, particularly in Section 3.4. This will enable readers to understand your research steps and technical approaches more intuitively.

4. Clarify the Use of Fundamental Theory:

You mention the application of fundamental theory in Chapter 3, but in Section 4.2, you state that it is not suitable. This inconsistency may confuse readers. Please clearly indicate whether you have utilized the fundamental theory. If not, explain the reasons and specify the alternative methods or theoretical basis you have adopted.

5. Provide Justification for the Selection of Label Words in Chapter 4; Conclusions Require Further Verification:

Due to the absence of relevant figures, I am skeptical about your selection of specific label words. The choice of these labels should be grounded in clear data or analytical results. Similarly, the final conclusions lack sufficient supporting evidence. I suggest providing related figures and detailed analyses to enhance the credibility of your conclusions.

6. Supplement the Logical Basis for the First Group Classification; Strengthen Logical Connections Between Sections:

There is a lack of clear logical reasoning and explanation for the first set of classifications in Section 4.2.1. Please provide additional clarification on the basis and criteria for your classification. Similarly, there is an absence of logical connection between Chapter 4 and Section 4.2.2. Enhancing the coherence between sections will ensure the paper's structural consistency and logical integrity.

7. Ensure References Strictly Follow the Journal's Guidelines; Standardize the Citation of Chinese Journals:

Please verify that the formatting of your references fully complies with the submission guidelines of the target journal. For citations of Chinese journals, determine whether to cite the original text directly or provide English translations or transliterations, as specified by the journal's guidelines. Ensuring uniformity and standardization in your references will enhance the professional quality of your paper.

8.Language Quality and Writing Style

The English expression in your manuscript has not yet reached the standard required for publication. There is a strong influence of Chinese language structures and expressions in the article, which adversely affects the clarity and readability of the text. I recommend that you undertake a comprehensive language polishing and revision of the manuscript to enhance the quality of your English writing. Seeking assistance from a professional English editor with academic writing experience or from native English speakers may help improve the language level of your manuscript.

Reviewer #5: First of all, I'd like to thank the authors for submitting their work to PLOS ONE. In this manuscript, the authors combine TF-IDF with grounded theory to analyze online public opinion in the context of a specific food safety incident, the "Rat Head and Duck Neck" case. This combination of methods is intended to identify public concerns and construct a theoretical model for understanding the dynamics of food safety-related online discourse. The topic is both relevant and timely, given the increasing societal reliance on social media platforms for public discourse.

Regretfully, as it stands, I cannot recommend the article for publication without at least undergoing a major revision. Below, I outline the major and minor issues with the work.

### **Major Issues**

1. **Reliability of Results**:

- The authors state in their conclusions: "In addition, the manual coding method is not scientific and objective enough, and it is necessary to code in a more scientific way in the future, and then construct a theoretical model based on the coding results."

- Such a statement raises serious concerns about the reliability of the results. While grounded theory inherently involves some degree of subjectivity due to its qualitative nature, the authors seem to acknowledge significant flaws in their application of this method. This acknowledgment undermines confidence in the findings and the constructed model.

- I suggest the authors clarify and reframe this statement. Instead of dismissing their coding method as "not scientific," they could highlight its limitations while justifying why this approach was chosen. Furthermore, they should explain what measures were taken to ensure rigor (e.g., inter-coder reliability, validation procedures) and why grounded theory is appropriate for the topic at hand.

- Alternatively, the authors should consider employing more systematic or software-assisted coding methods to strengthen the objectivity and reproducibility of their findings.

2. **Lack of Citation and Discussion of Related Work**:

- A brief literature search on this topic revealed two important references:

1. Liu, J., Wang, S., Wang, Z., & Chen, S. (2024). Research on online public opinion dissemination and emergency countermeasures of food safety in universities—take the rat head and duck neck incident in China as an example. _Frontiers in Public Health, 11_, 1346577.

2. Ma, B., & Zheng, R. (2024). Exploring Food Safety Emergency Incidents on Sina Weibo: Using Text Mining and Sentiment Evolution. _Journal of Food Protection_, 100418.

- While the second paper by Ma and Zheng is very recent, and it is entirely understandable that it might have been missed if the authors submitted their work earlier, it is essential to consider its findings in a revised version. The first paper, however, has been available for a longer time, and it directly addresses the same incident using a "life cycle" framework to analyze public opinion dissemination. Missing this reference is more difficult to justify.

- Both papers analyze the same food safety incident, and their methodologies overlap or even exceed those used in the paper under review. For instance:

- Liu et al. propose a comprehensive "life cycle" framework for public opinion dissemination, offering a detailed thematic analysis and emergency countermeasures.

- Ma and Zheng employ advanced NLP methods (e.g., BERT-TextCNN and BERTopic) and network analysis to analyze sentiment dynamics and information dissemination.

- I strongly encourage the authors to include both references in a potential revision and actively compare and contrast their findings with these studies. Specifically, the authors should:

- Highlight what their integration of grounded theory and text mining contributes beyond the existing studies.

- Explain how their constructed model differs in terms of methodology or practical application.

- Acknowledge the strengths of these studies while situating their work within the broader literature on food safety incidents.

### **Minor Issues**

1. **Literature Review**:

- The literature review would benefit from a brief classification of the main families of approaches to online public opinion analysis.

- Additionally, the authors should clearly position their methodology within these categories, emphasizing what their chosen approach (grounded theory combined with text mining) uniquely brings to the table.

2. **Explanation of LDA and Pipeline**:

- The description of LDA in the methodology section could be clearer. While the figure illustrating the process is helpful, the accompanying text should provide a step-by-step explanation.

- The authors should also provide a more detailed narrative of their workflow, from data preprocessing to model construction, to enhance reproducibility.

3. **Policy Recommendations**:

- The recommendations are practical but generic. The authors could strengthen this section by linking their suggestions more explicitly to the categories and themes identified in their model.

### **Final Thoughts**

This paper addresses a significant and timely topic. However, the issues outlined above—particularly the concerns about reliability, lack of engagement with related literature—must be addressed before the manuscript can be considered for publication. A major revision should focus on:

- Including and discussing relevant prior work, clearly articulating the novelty and relevance of the authors' contributions.

- Strengthening the rigor of the coding process or justifying the use of grounded theory in this context.

- Clarifying the methodology and improving the presentation of results and recommendations.

Thank you for the opportunity to review this work. I hope my comments assist the authors in further refining their study.

Reviewer #6: This study analyzes public opinion formation regarding the "Rat Head Duck Neck Incident" in China by applying text analysis to Weibo data. It addresses an important topic concerning public opinion on food safety. However, there are areas in the analysis and presentation that need improvement. Therefore, a major revision is recommended.

[Major Revisions]

1. The content presented in Figure 1 is unclear. If it aims to show the steps in data analysis, provide specific descriptions in analysis methods.

2. Referring to a co-occurrence network of words as a "social network" is inappropriate such as in Figure 5. "Social networks" refer to networks between individuals. Please use proper terminology.

3. Clarify the relationship between 6 topics and words in Table 3. Specify how certain words support the interpretations made in the discussion. Provide explicit explanations.

4. The procedures for grounded theory analysis are unclear. Add detailed steps and a diagram in Section 3.3 to explain the methodology more clearly.

5. While Section 4 discusses responses to this incident within the context of China, a broader discussion is required for an academic paper. Generalize the findings and implications of this study to provide insights into public opinion formation on food safety.

[Minor Revisions]

6. The related works in Section 2 is too long and mentions models like SIR, which are not directly related to this study. I recommend that summarize this section concisely and instead focus on prior studies using natural language processing or grounded theory.

7. Are "Network Public Opinion" and "Online Public Opinion" synonymous? If so, it would be clearer to use "Online Public Opinion" consistently throughout the paper.

**Do you want your identity to be public for this peer review?** For information about this choice, including consent withdrawal, please see our Privacy Policy

Reviewer #1: No

Reviewer #2: No

Reviewer #3: No

Reviewer #4: No

Reviewer #5: No

Reviewer #6: **Yes: ** Masaki Chujyo

---

## [Author Response · Author response to Decision Letter 1]

24 Feb 2025

We are deeply grateful for your thorough and insightful review of our article. Following your valuable suggestions, we have made extensive revisions to our initial draft. The detailed corrections and improvements are outlined below.

Issues raised by editor regarding data availability statements and funds have been revised.

I have made the following changes to the suggestions made by the first reviewer:

With regard to the suggested changes to the literature review section, we have revised the literature review section by reorganizing the content to review previous studies in a more analytical way, rather than listing the studies in chronological order. The literature review is now more critical and comprehensive and clearly identifies gaps in current knowledge as well as the unique contributions of this study.

Regarding the section on methodology, it has been appropriately streamlined in the revised text, while regarding the identification of the number of topics, the reasons for choosing perplexity as an assessment metric for topic modeling and the advantages of perplexity over other options have been clearly stated in the paper.

Regarding the section on deepening the discussion of the analysis, in response to the reviewers’ suggestions, a detailed explanation of the transition has been provided before proceeding with the coding of the grounded theory to ensure the depth and coherence of the analysis.

In the section on strengthening practical implications, specific insights have been added based on the analysis of the paper and the summary and recommendations section at the end of the article has been revised to enhance the practical value and flow of the paper.

I have made the following changes to the suggestions made by the second reviewer:

The writing of the literature review has been optimized based on the reviewers’ suggestions by using clearer, simpler sentences and using technical terms only when necessary to improve readability and comprehension.

I have made the following changes to the suggestions made by the third reviewer:

On the issue of social media posts, a discussion of the ethics of using social media posts has been added to the data collection section, and transparency has been ensured on this.

Regarding the issue of deleting emojis, this has been explained in the preprocessing section of the revised text.

Regarding the merging of synonyms in the word cloud section, the merging was done manually as only the top 100 words in terms of frequency were selected for this paper. The definition of co-occurring phrases in this paper is phrases that co-occur in a single comment, and as for the semantic relationship network Figure in the paper, it has also been drawn based on the frequency of co-occurrence of words, and has been based on the reviewer’s suggestion of using a more active tense and the first person in the paper to describe the different aspects of the study in order to show how the study was conducted in a clearer way.

The range of the number of topics chosen in the paper for the confusion figure is 0-15, 15 is not a fixed value, just make sure that the optimum number of topics is between the ranges but it should not be chosen to be too large or too small; also the paper has not chosen any other method to determine the number of topics for the reasons explained in the perplexity section of the revised paper.

The advantages and disadvantages of the methods used in this paper have been discussed in the summary section at the end of the revised text.

The explanation from spatial coding to axial coding has been reflected in the revised text: the initial categories are generalized and enhanced by combining the coding results of previous researchers. The similarities and differences between the theoretical memos in traditional grounded theory and the methods we used in our study have been discussed in detail in the model building section of the revised text.

Regarding the reviewer’s suggestion to use the COREQ framework to evaluate this paper: first of all, we are very grateful to the reviewer for this valuable suggestion, and since it is not easy to list all the 32 entries in this paper, this paper only lists some of the core entries such as the research questions, the data collection methods, and the data analyses, which have been interspersed with the various panels in the text.

I have made the following changes to the suggestions made by the fourth reviewer:

All figures and tables have been added in the revised text to support our analysis and conclusions.

In response to the comments made in the literature review section, we have added critical discussions and unique insights in the related work section as suggested and broken it down by research focus to improve readability and logic. We have also streamlined and optimized the lengthy content.

As suggested by the reviewers, we have simplified the theoretical descriptions on methodology and highlighted specific research steps and data processing methods

We thank the reviewers for their corrections and we have noted the inconsistency between the two. and have explained this in the revised text: the traditional grounded theory is not suitable for this paper, which is a combination of text mining and grounded theory.

The choice of tag words is determined by the topic words and the words with higher TF-IDF values. The six topics under the topic words in the text have been selected ten in Table 3 (of course, this is only part of the words under each topic), TF-IDF value of the highest ten words in the text is also listed in Figure 5, of course, the above is only a part of the tag words, the rest of the tag words are also selected from the topic words or words with high TF-IDF values. As the selection of tag words reaches 90, it is unrealistic to list all the tag words in the table of topic words and the figure of words with high TF-IDF value.

Regarding the reviewer’s suggestion about the lack of explicit logical reasoning and explanation in the coding process of grounded theory, it has been explained and modified in the revised text to ensure the consistency of the paper’s structure and logical integrity.

Thanks to the reviewers’ suggestions, we will check and adjust the reference formatting in strict accordance with the submission guidelines of the target journal to enhance the professional quality of the paper.

We will make linguistic touches and revisions to the paper as suggested to ensure that the clarity and readability of the text meets publication standards.

I have made the following changes to the suggestions made by the fifth reviewer:

Based on the reviewers’ suggestions, revisions have been made in the text: the limitations of traditional grounded theory have been emphasized, as well as the reasons for combining text mining and grounded theory in this paper, and in order to ensure the rigor of the coding, the process of coding has also been exchanged with some of the students and the coding results of previous researchers have been referred to.

Regarding the two references mentioned in the reviewers’ revisions, they have been cited and compared in the revised text, mentioning the strengths and weaknesses of these studies.

According to the reviewer’s suggestion, a categorization has been made in the literature review section for the different research methods on network public opinion and the unique features of the research methodology of this paper have been emphasized.

Changes have been made to the introduction of the LDA topic model in the revised text, and the entire research process has also been refined to improve reproducibility.

The categories identified in the model and the results of the analysis have been used as a basis for policy recommendations based on the reviewers’ revisions.

I have made the following changes to the suggestions made by the sixth reviewer:

Figure 1 presents mainly the technical roadmap of the paper, aiming to roughly show the research steps of this paper.

Thanks to the reviewer’s suggestion, the name of Figure 5 has been changed to Semantic Relationship Network Figure

As suggested by the reviewers, it has been specified how these subject terms under each topic support our explanations in the discussion.

Thanks to the reviewer’s suggestion, we have added a clear figure in 3.3 to explain grounded theory more clearly.

Following the reviewer’s suggestion, we will expand on the discussion in Section 4 by broadening the findings and implications to provide broader insights into public opinion formation on food safety.

We thank the reviewers for their suggestions for the literature review section, which has been revised in accordance with the revisions.

With regard to “network public opinion” and “online public opinion”, this has been harmonized in the revised text.

---

## [Decision Letter · Decision Letter 1]

Dear Dr. LIN,

Thank you for submitting your manuscript to PLOS ONE. After careful consideration, we feel that it has merit but does not fully meet PLOS ONE’s publication criteria as it currently stands. Therefore, we invite you to submit a revised version of the manuscript that addresses the points raised during the review process.

We look forward to receiving your revised manuscript.

Kind regards,

Tinggui Chen

Academic Editor

PLOS ONE

Additional Editor Comments:

I have completed my evaluation of your manuscript. The reviewers recommend reconsideration of your manuscript following major revision. I invite you to resubmit your manuscript after addressing the comments below.

Reviewers' comments:

Reviewer's Responses to Questions

**Comments to the Author**

Reviewer #2: All comments have been addressed

Reviewer #3: All comments have been addressed

Reviewer #5: All comments have been addressed

Reviewer #6: (No Response)

2. Is the manuscript technically sound, and do the data support the conclusions?

Reviewer #2: Yes

Reviewer #3: Yes

Reviewer #5: Yes

Reviewer #6: Partly

3. Has the statistical analysis been performed appropriately and rigorously?

Reviewer #2: Yes

Reviewer #3: Yes

Reviewer #5: Yes

Reviewer #6: N/A

4. Have the authors made all data underlying the findings in their manuscript fully available?

Reviewer #2: Yes

Reviewer #3: Yes

Reviewer #5: Yes

Reviewer #6: Yes

5. Is the manuscript presented in an intelligible fashion and written in standard English?

Reviewer #2: Yes

Reviewer #3: Yes

Reviewer #5: Yes

Reviewer #6: Yes

Reviewer #2: There are massive improvement in this article for the literature review section, which has a strong justification

With regard to “network public opinion” and “online public opinion”, this has been harmonized in significant contribution to the article.

Reviewer #3: Thank your for your substantial revisions in light of the feedback from all the reviewers. I think the paper is now much clearer and explains your methodological approach and steps taken to ensure rigour. I think interlinking your processes to manual grounded theory is very useful for other researchers and I appreciate the use of some COREQ items whilst appreciating the challenges of doing so.

Reviewer #5: I appreciate the authors' efforts to amend all the suggestions and changes proposed to improve the manuscript. After this extensive round of reviewing, I consider it apt for publication

Reviewer #6: The authors have responded to the reviewers' questions, but it is unclear exactly how they have rewritten the text. I consider this to be unsuitable for peer review, and recommend rejection.

**Do you want your identity to be public for this peer review?** For information about this choice, including consent withdrawal, please see our Privacy Policy

Reviewer #2: No

Reviewer #3: No

Reviewer #5: No

Reviewer #6: No

---

## [Author Response · Author response to Decision Letter 2]

19 Mar 2025

We are deeply grateful to the reviewers for their thorough and insightful comments on our manuscript. Based on your invaluable suggestions, we have undertaken another round of revisions and refinements. The detailed corrections and improvements are outlined below.

I have made the following changes to the suggestions made by the second reviewer:

Thank you very much for your comments on the revision of the literature review section in the previous round of revisions, and thank you for recognizing our manuscript.

I have made the following changes to the suggestions made by the third reviewer:

Thank you very much for your valuable comments during the last round of revisions, indeed it is one of the highlights of this paper. At the same time, COREQ has a total of 32 entries, it is indeed unrealistic to use them completely in this paper, so only some core entries have been selected in this paper, thank you for your understanding.

I have made the following changes to the suggestions made by the fifth reviewer:

Thank you very much for the revisions you made during the last round of revisions, and thank you for recognizing our revision process and the revised manuscript.

I have made the following changes to the suggestions made by the sixth reviewer:

About how we rewrite the text.

Figure 1 would have presented the technology roadmap for this paper, but given the short length of this paper, it was decided to remove it.

Thanks to the reviewer’s suggestion, the name of Figure 5 has been changed to Semantic Relationship Network Figure

As suggested by the reviewer, we have shown how those subject terms under each theme support our explanations in the discussion (including in the context of text specifics).

Thanks to the reviewer’s suggestion, we have added a clear figure in Method section (Figure 2) to explain grounded theory more clearly.

Following the reviewer’s suggestion, we will expand on the discussion in Section 4 by broadening the findings and implications to provide broader insights into public opinion formation on food safety.

We thank the reviewers for their suggestions for the literature review section, which has been revised in accordance with the revisions.

With regard to “network public opinion” and “online public opinion”, this has been harmonized in the revised text.

---

## [Decision Letter · Decision Letter 2]

Research on the generation of food safety network public opinion - taking the Chinese Mouse Head Duck Neck Incident as an example

PONE-D-24-39624R2

Dear Dr. LIN,

We’re pleased to inform you that your manuscript has been judged scientifically suitable for publication and will be formally accepted for publication once it meets all outstanding technical requirements.

Kind regards,

Tinggui Chen

Academic Editor

PLOS ONE

Additional Editor Comments (optional):

Reviewers' comments:

Reviewer's Responses to Questions

**Comments to the Author**

Reviewer #2: All comments have been addressed

Reviewer #3: All comments have been addressed

2. Is the manuscript technically sound, and do the data support the conclusions?

Reviewer #2: Yes

Reviewer #3: Yes

3. Has the statistical analysis been performed appropriately and rigorously?

Reviewer #2: Yes

Reviewer #3: Yes

4. Have the authors made all data underlying the findings in their manuscript fully available?

Reviewer #2: Yes

Reviewer #3: Yes

5. Is the manuscript presented in an intelligible fashion and written in standard English?

Reviewer #2: Yes

Reviewer #3: Yes

Reviewer #2: This paper examines a food safety issue at a university in Jiangxi Province, using text mining and grounded theory to study network public opinion. First, text mining is used to discover public issues during public opinion periods. This determines topic words and keywords. High TF-IDF values are used for grounded theoretical coding. We developed a mechanism to generate public opinion on food safety issues in networks. Analysis of the model revealed four key contributors in the construction of network public opinion: the subject, intermediary, object, and government authority. Targeted suggestions were made from three perspectives: public opinion subject, intermediary, and government authority. The research has flaws. The selected food safety incidents from campus incidents may not be comprehensive enough. Future research should include more comprehensive incidents.

Reviewer #3: (No Response)

**Do you want your identity to be public for this peer review?** For information about this choice, including consent withdrawal, please see our Privacy Policy

Reviewer #2: No

Reviewer #3: No

---

## [Editor Report · Acceptance letter]

PONE-D-24-39624R2

PLOS ONE

Dear Dr. LIN,

I'm pleased to inform you that your manuscript has been deemed suitable for publication in PLOS ONE. Congratulations! Your manuscript is now being handed over to our production team.

Kind regards,

on behalf of

Dr. Tinggui Chen

Academic Editor

PLOS ONE